# Stress Buffering and Longevity Effects of Amber Extract on *Caenorhabditis elegans* (*C. elegans*)

**DOI:** 10.3390/molecules27123858

**Published:** 2022-06-16

**Authors:** Sandra Somuah-Asante, Kazuichi Sakamoto

**Affiliations:** 1Tsukuba Life Science Innovation Program (T-LSI), University of Tsukuba, 1-1-1 Tennodai, Tsukuba 305-8577, Japan; ssasante1@gmail.com; 2Faculty of Life and Environmental Sciences, University of Tsukuba, 1-1-1 Tennodai, Tsukuba 305-8577, Japan

**Keywords:** *C. elegans*, amber extract, anti-stress, antiaging, antioxidant, stress hormones, DAF-16

## Abstract

Amber is a fossilized tree resin historically used in wound healing and stress relief. Unfortunately, there is no concrete scientific evidence supporting such efficacy. Here, the stress buffering and longevity effect of Amber extract (AE) in *Caenorhabditis elegans* (*C. elegans*) was investigated. Survival assays, health span assays, Enzyme-Linked Immunosorbent Assay (ELISA), Stress biomarker detection assays, Green Fluorescence Proteins (GFP), Real Time quantitative PCR (RT-qPCR) and *C. elegans* mutants were employed to investigate the stress buffering and longevity effect of AE. In the study, it was observed that AE supplementation improved health span and survival in both normal and stressed worms. Additionally, AE positively regulated stress hormones (cortisol, oxytocin, and dopamine) and decreased fat and reactive oxygen species (ROS) accumulation. Through the Insulin/IGF-1 signaling (IIS) pathway, AE enhanced the nuclear localization of DAF-16 and the expression of heat shock proteins and antioxidant genes in GFP-tagged worms and at messenger RNA levels. Finally, AE failed to increase the survival of *daf-16*, *daf-2*, *skn-1* and *hsf-1* loss-of-function mutants, confirming the involvement of the IIS pathway. Evidently, AE supplementation relieves stress and enhances longevity. Thus, amber may be a potent nutraceutical for stress relief.

## 1. Introduction

Stress is any physical or psychological threat to an organism that may be triggered by environmental or psychological changes. The length or persistence of stress can have a great impact on an individual’s wellbeing [1]. Generally, a short moment of stress (acute stress) to overcome danger, like in a fight or flight scenario, is helpful as a protective mechanism. However, a prolonged exposure to stress can negatively affect quality life and long life. Prolonged stress exposure (chronic stress) can cause hormone and neurotransmitter levels to become erratic, leading to headaches, fatigue, depression, obesity, aging and aging-related diseases and many more complications [2]. Stress also causes the accumulation of reactive oxygen species (ROS) which damage cells, leading to cell death and aging [3]. The endocrine and nervous system, linked by the Hypothalamic Pituitary Adrenal (HPA) axis, have been implicated to regulate chemical signaling to maintain homeostasis [4]. Although long life is of prime interest, the quality of such long life is also important. Therefore, quality of life, herein referred to as health span, is characterized by a life free from diseases [5]. Unfortunately, this (health span) is mostly challenged by stress. Although exercises, meditation, sleep etc. are recommended for managing stress [6], these strategies seem to be inefficient in this fast-paced world. Therefore, it is advisable to explore effective ways of managing stress and reducing stress or lifestyle-induced complications.

Plants over the years have been of great importance to living organisms as a source of food, shelter, and medicine. Several natural products of plant origin including Emodin [7], *Momordica charantia* [8], *Moringa oleifera* [9] and many others, have been reported to increase stress tolerance and extend lifespan. Other nutraceuticals like Ginseng, Gingko biloba, Valerian etc., have also been highlighted to regulate hormone and neurotransmitter levels which aid in reducing stress-related depression and anxiety [10]. One potential understudied nutraceutical is amber, commonly found on the pine tree. Amber is a brownish gum-like tree resin from the Sciadopityaceae family that has fossilized over time. It contains succinic acid [11] and many phytochemicals including terpenoids [12,13]. Terpenoids have been highlighted to possess an antioxidant ability and offer protection against oxidative stress-induced diseases such as cancers, diabetes, neurodegenerative diseases and many others [14]. The diterpenoids agathic acid, primaric acid and abietic acid have been discovered as the major components of Baltic Amber [15,16]. Historically, herbal medicine practitioners used amber extract in various ailments including wound healing and stress relief. However, in modern times, amber has gained popularity in the fashion industry for making jewelry. Though at present it is mostly used for decorative purposes, people still believe that wearing an amber necklace or bracelet can relieve stress, pain and induce relaxation; a particular reason for its use as teethers in babies [17]. There have been claims amongst amber jewelry merchants that the succinic acid in amber, specifically Baltic Amber, may be responsible for its soothing effect [17]. However, Nissen et al., 2019, have shown that this claim is likely not possible based on the release and absorption mechanism of succinic acid [18]. To our knowledge, there is not yet scientific evidence for the stress relieving effect of AE on *C. elegans*. Recently, our research group reported several other benefits of AE in the quest to investigate folkloric beliefs pertaining to its usage. First, Tian et al., 2021, discovered that AE possessed anti-inflammatory properties [19] whilst Luo et al., 2021, reported that AE may be potent in managing Alzheimer’s disease [20]. Additionally, Sogo et al., 2021, discovered that AE could reduce fat accumulation [12]. These cell-based studies together have shown that amber contains potent phytochemicals with various health benefits worth investigating.

*C. elegans* used in this study are a simple but useful model organism that have provided useful insights into longevity, stress resistance and their associated molecular pathways [21]. Their hermaphroditic nature and short life span enable a quick turnover, making them easy to culture in the lab [22]. Additionally, they possess a transparent body which enables clear visualization during experimentations, especially with Green Fluorescence Protein (GFP) tagged worms. Most importantly, *C. elegans* possess a genetic make-up like that of higher organisms hence their studies are highly translatable to that of humans [22].

In this study, we investigated the effects of AE on the survival of *C. elegans* under normal conditions and when exposed to stressors. Since we are the first to study the effects of AE on *C. elegans*, it was important to establish, through research, that AE does not pose any harm in the development and physiological functioning of the worms. We again studied the effect of AE on the regulation of stress biomarkers; hormones/neurotransmitter levels, ROS, and fat accumulation. The expression of longevity and stress regulatory genes were also monitored with GFP-fused reporters and at mRNA levels. Molecular pathway(s) that may be involved in AE-mediated activity were determined using loss-of-function *C. elegans* strains (mutants).

## 2. Results

### 2.1. AE Increased Lifespan and Improved Health Span in C. elegans

First, the effect of AE on the lifespan of N2 worms was investigated by determining the survival of AE-treated (5, 25 and 50 µg/mL) and non-treated worms. As shown in Figure 1A, the survival rate of AE-treated worms significantly increased compared to the control group, 5% DMSO. Additionally, the survival of worms in the control group was compared to that of worms fed with *E. coli* OP50—only to ascertain if there were any notable effects from the use of DMSO on the worms. No significant differences were observed in the survival of worms treated with DMSO and *E. coli* OP50 food (Figure 1A, Appendix A). Some researchers have claimed that longevity can be achieved with dietary restrictions [23]. Hence, the possibility of AE-treated worms starving to achieve longevity was investigated in addition to whether treatment with AE posed any physical damages or changes to the worms. Here, reproduction capacity, body length and motility determination were employed as indicators for health span. It was found in Figure 1B that AE supplementation did not significantly change both the daily and the overall fertility of worms. In the body length determination (Figure 1C), it was also observed that worms fed well with no physical changes in length just like the control group. In Figure 1D, 25 µg/mL and 50 µg/mL AE treatment enhanced muscle movement as worms aged: day 6 and day 6 and 9 respectively. On the contrary, this effect was not seen in 5 µg/mL AE-treated and control worms. Based on these effects, AE concentration of 50 µg/mL was selected as the most consistent and effective concentration for further analysis.

### 2.2. AE Enhanced Stress Tolerance in C. elegans

Stressors take various forms; heat and oxidative stress are the most investigated stressors in *C. elegans* studies [21]. Here, the protective effect of AE against heat and oxidative stress in N2 worms were investigated. Exposure of worms to heat at 35 °C for 7 h resulted in a significant increase in the survival of AE-treated worms compared to the non-treated group as shown in Figure 2A, Appendix A. When N2 worms were exposed to 0.1% hydrogen peroxide oxidative stress, a significant increase in stress tolerance but not overall lifespan was observed in AE-treated worms (Figure 2B, Appendix A). This suggests that AE has potential to alleviate stress.

### 2.3. AE Positively Regulated Levels of Stress Biomarkers in C. elegans

When an organism is exposed to stress, several hormones, neurotransmitters and other biological markers become erratic. Thus, this affects normal physiological functioning which can lead to stress-induced diseases and disorders [24,25]. To determine how AE regulates stress biomarkers, the levels of stress hormones, cortisol, oxytocin and dopamine, in addition to fat and ROS accumulation in *C. elegans* were investigated. In Figure 3A, the low production of cortisol (Endo) by the worms depicted that, endogenously, *C. elegans* (~3000 worms) were unable to produce enough cortisol for analysis using ELISA. Hence, the addition of 17.5 mM Cortisone. Cortisone is an inactive substrate which is converted to cortisol through the enzymatic activities of the 11β-HSD 1 enzyme [26]. It was observed, after additional cortisone treatment in Exo (*E. coli* OP50 fed worms; positive control), DMSO and 50 µg/mL groups that had AE treatment had significantly reduced cortisol levels compared to the control groups (Exo and DMSO). Thus, this suggests that AE may possess the ability to inhibit the activities of the 11β-HSD 1 enzyme. Interestingly, as shown in Figure 3B,C, AE treatment significantly increased oxytocin and dopamine levels, respectively. This also suggests that AE is essential to increase social bonding, interactions and pleasure. A common stress associated disease/disorder is obesity, characterized as the excessive accumulation of fat due to irregular eating patterns caused by stress [27]. In Figure 3D, it was observed that AE treatment significantly reduced fat in worms compared to the control group, depicting that AE may be effective in controlling obesity. Additionally, in Figure 3E, AE supplementation in worms showed a significant reduction in the accumulation of intracellular ROS. However, AE was ineffective in scavenging ROS in vitro (Figure 3F). Thus, this implies that AE possibly does not possess inherent antioxidant activity.

### 2.4. AE Increased the Nuclear Localization of DAF-16 and Expression of HSP-16.2, SOD-3 and GST-4

To gain insights into the molecular mechanism(s) that may be involved in the AE-mediated stress buffering and longevity effects, expression of DAF-16, a common protein associated with longevity, was first studied. DAF-16 is the human ortholog of the Forkhead box O (FOXO) family and a key transcription factor involved in several biological activities including longevity and stress resistance [28]. In the nucleus, DAF-16 on its own, or in conjunction with other transcription factors such as the heat shock factor-1 (HSF-1) and skinhead-1 (SKN-1), coordinates the expression of several stress responsive genes such as superoxide dismutase, sod-3 [29,30], glutathione s-transferase, gst-4 [31], catalase, ctl-1 [32] and heat shock proteins, hsp-16.2 [33]. One important function for a drug under longevity and stress resistance screening is to be able to activate DAF-16 for its longevity and stress resistance activity. To determine the involvement of DAF-16, the subcellular localization of DAF-16 and the expression of its target genes were studied. In Figure 4A, an increase in the nuclear localization of DAF-16::GFP was observed in TJ365 worms exposed to AE treatment. Additionally, the expressions of heat shock proteins, HSP-16.2 (Figure 4B) and antioxidants genes, SOD-3 (Figure 4C) and GST-4 (Figure 4D), were enhanced with AE supplementation compared to the control.

### 2.5. AE Increased the Messenger RNA Expression of daf-16 and Its Target Genes

Additionally, real time quantitative PCR analysis was conducted to confirm the expression of *daf-16* and its target genes that may be related to the activity of AE. As shown in Figure 5, a significant increase in the expressions of *daf-16*, *hsp-16.2*, *hsp-70*, *sod-3*, *gst-4* and *ctl-1* in AE-treated worms was observed. Thus, this confirms that *daf-16* and its target genes (gene sequence in Appendix A) contributed to the stress resistance and longevity effects of AE.

### 2.6. AE-Mediated Stress Tolerance and Longevity Activity May Be via the IIS Pathway

To confirm that the IIS pathway plays a role in the benefits of AE, the oxidative stress tolerance of *daf-2*, *daf-16*, *skn-1* and *hsf-1* mutants was investigated. It was observed in Figure 6A that AE-treated *daf-2* (*e1370*) mutants failed to survive oxidative stress longer than the control group. Although, both control and AE-treated groups exhibited prolonged tolerance to oxidative stress compared to N2 worms. Additionally, *daf-16* (*mgDf50*) mutants lost their stress tolerating ability even when supplemented with AE (Figure 6B). Due to the upregulation of *gst-4* observed in Figure 4D and Figure 5, *skn-1*, an upstream transcription factor of *gst-4*, was studied. *Skn-1* is an ortholog of the human Nuclear factor-erythroid 2-related factor 2 (*Nrf 2*) and is involved in detoxification and antioxidant activities particularly in oxidative stress [34,35]. In Figure 6C, it was also observed that AE supplementation failed to ensure oxidative stress tolerance in *skn-1* (*tm4241*) mutants. *Hsf-1* (*sy441*) mutants, when exposed to AE treatment, also did not show any significant tolerance to oxidative stress (Figure 6D). This all implies that these genes are related, and AE depends on its activities in the IIS pathway for stress tolerance and longevity.

## 3. Discussion

Stress-induced complications and mortality are rapidly increasing in the human population due to the demands of this fast-paced world. This has become a global concern as lifestyle is a major contributor to this menace. Work life balance is almost impossible to achieve in modern times. Hence, dietary supplementation with nutraceuticals with fewer side effects seems to be a positive way to maintain a healthy lifestyle.

In this study, the stress buffering and lifespan extension effects of amber, a promising nutraceutical, on *C. elegans* were explored. The increased lifespan of worms exposed to AE is an indication that AE may possess potent phytochemicals with anti-aging properties. Researchers have shown that several crude extracts including *Caesalpinia mimosoides*, *Anacardium occidentale* and *Moringa oleifera* demonstrated the ability to extend lifespan in *C. elegans* [9,29,30]. These extracts generally improve health span. Here, AE improved the health span of worms by maintaining the fertility and normal growth (body length) of worms, without any physical defects or developmental changes. A few studies have suggested that a ‘trade off’ mechanism exists between the ability to reproduce and live long, in that worms can give up reproduction to live longer or vice versa [36,37]. Conversely, it is evident that this concept is not a general rule as worms supplemented with AE lived long and maintained normal reproduction capacity in our study. A similar observation was made by Lin et al., 2019, when rosmarinic acid prolonged lifespan and maintained *C. elegans* reproduction as well [38]. In *C. elegans* studies, motility is employed as a biomarker to determine improved well-being and lifespan [39]. This implies that the ability to maintain an increased muscle movement is a projection for longevity while a loss of muscle function is an indication of age decline. Thus, the increased motility observed with AE supplementation confirmed the ability of AE to prolong lifespan. Amber, containing agathic acid, primaric acid and abietic acid, has been reported to possess antibacterial activity against *E. coli* and other bacteria [16]. Perhaps this antibacterial activity of amber contributed to lifespan extension of worms by inhibiting the proliferation and accumulation of live *E. coli* in worms’ guts that could negatively affect health span and lifespan.

In living organisms, the prevalence of stress is known to cause ROS accumulation [39] and hormonal imbalance [25] which negatively affect the quality of life and shortens lifespan. In our study, it was observed that AE scavenged intracellular ROS in vivo but failed to exhibit in vitro ROS scavenging abilities. According to Shahidi et al., 2015, concentration, temperature, synergism and many other factors can affect the productivity of antioxidants in vitro [40]. Due to this, it was proposed that one or several of these factors may have contributed to the differential observation in the in vivo and in vitro antioxidant activity of AE. Perhaps, the concentration of AE (50 µg/mL) was too low to exert inherent antioxidant abilities. Though, it may be enough to upregulate and synergistically act with antioxidant genes and enzymes in vivo for an antioxidant effect. This possibility was observed when AE-treated worms showed an upregulation of antioxidant genes, *sod-3* and *gst-4* in GFP-tagged worms and at mRNA levels, and *ctl-1* at mRNA levels. These findings somehow explain the protective effect of AE against hydrogen peroxide-induced oxidative stress. Heat shock proteins (hsps) protect cells from protein misfolding and damage during stress and are known to be strongly related to stress tolerance and longevity [5]. The activation of most hsps is induced by the transcription factor, heat shock factor 1 (HSF-1) [41]. Thus, the upregulation of *hsp-16.2* in both GFP-tagged worms and at mRNA level and *hsp-70* at mRNA level may have contributed to the AE-mediated stress survival and longevity. The failure of AE-treated *hsf-1* mutants to survive oxidative stress longer than non-treated worms depicted that AE requires *hsf-1* to mediate its stress tolerance effects.

Previous studies have reported that the endocrine system, specifically hormone regulation, is involved in the lifespan determination of *C. elegans* [39]. The ability of AE to decrease cortisol and increase both oxytocin and dopamine levels is a useful intervention that may have contributed to promoting longevity in *C. elegans*. In vertebrates, the hypothalamus-Pituitary-Adrenal (HPA) axis is activated during stress to control stress hormone levels to maintain homeostasis [42]. Cortisol, a glucocorticoid in the HPA axis, is involved in many homeostatic interventions during stress. It is a stress hormone produced mainly from cholesterol and intracellularly, it is produced through the 11 beta hydroxysteroid dehydrogenase (11β-HSD) 1 pathway [43]. Excessive cortisol production, common in chronic stress, has been shown to lead to several metabolic disorders [44] including central obesity [43]. High cortisol levels hinder glucose uptake and stimulate appetite, thereby causing an organism to eat more and resulting in central obesity [26,45]. Currently, metabolic syndrome researchers are interested in compounds that can inhibit the 11β-HSD1 enzyme as effective therapeutic agents to tackle endocrine disorders like obesity and diabetes [45]. In view of that, it is speculated that AE may be an inhibitor of the 11β-HSD1 enzyme, although further studies are required. The reduction in cortisol levels was consistent with the reduced fat accumulation observed in this study, although the exact mechanism is unknown. Interestingly, an increase in oxidative stress has been reported to cause an increase in cortisol release [46]. Suggesting that, an ability to reduce oxidative stress may imply an ability to reduce cortisol release. This was evident in the oxidative stress resistance and cortisol reduction ability of AE. Additionally, Ozbay et al., 2008 and McQuaid et al., 2016 have highlighted that stress resistance can be achieved through an increase in social support and interventions that seek to positively regulate hormone and neurotransmitter levels [47,48]. A neurotransmitter known to be present during social support and interventions like a simple hug, kiss, being a support system for others or receiving support from others is oxytocin. Oxytocin is a neuropeptide which increases with social bonding and is needed to exert positive effects on human behavior. It is known to have a direct correlation to social support and interactions where an increase in these activities is associated with increased oxytocin levels [49]. The increased oxytocin levels observed in our study depict that AE can maintain physical and mental wellbeing by modulating oxytocin levels in humans to improve social interactions and prevent stress-induced anxiety and depression. Chronic stress can make humans feel inadequate and unmotivated. Dopamine, a neuromodulator involved in the reward system, plays a role in motivating and encouraging a positive approach to life as a coping mechanism to stress [50]. In this study, AE increased dopamine levels in worms; a significant effect which suggests that AE may have potential in regulating dopamine levels in higher primates. The balancing (buffering) of these hormones and neurotransmitters caused by AE also suggests that AE, in addition to reducing stress and enhancing longevity, may exert antidepressant effects.

To determine the molecular pathway(s) by which AE may have exerted stress tolerance and longevity effects, it was suspected that the Insulin/Insulin-like growth factor 1 Signaling (IIS) pathway, through the activities of the transcription factor, DAF-16, was involved [51]. The IIS pathway is the most common molecular pathway exhibited by *C. elegans* and is conserved across many species for longevity and stress resistance. In this pathway, the key regulators are DAF-2, AGE-1 and DAF-16 [33]. Activation of the DAF-2 receptor initiates a phosphorylation cascade which leads to the phosphorylation of DAF-16. This inactivates DAF-16, causing it to remain in the cytoplasm [52]. However, upon the suppression of DAF-2, DAF-16 is activated and moves from the cytoplasm to the nucleus, where, on its own or in conjunction with other proteins like the HSF-1 and SKN-1, it upregulates several downstream genes [53]. Indeed, DAF-16 is the key transcription factor involved in the AE-mediated stress resistance and longevity activities, as it was observed that *daf-16* and its target genes *hsp-16.2*, *hsp-70*, *sod-3*, *gst-4* and *ctl-1* were all upregulated with AE treatment. More importantly, since *daf-2* and *daf-16* mutants completely lost their ability to tolerate oxidative stress with AE treatment, the IIS pathway was strongly presumed to be involved. Furthermore, according to Tullet et al., 2008, the IIS pathway can suppress both DAF-16 and SKN-1 [54], which conversely implies that suppressing the IIS pathway can activate both DAF-16 and SKN-1. This is deemed consistent with the loss of lifespan extension in *skn-1* mutants with AE supplementation in our findings. Clearly, AE depended on the health promoting activities of DAF-16 as well as SKN-1. Thus, it was proposed that AE exhibited its stress resistance and longevity effects at the molecular level via the collective actions of DAF-16, HSF-1 and SKN-1 in the IIS pathway.

A few limitations observed in this study pertain first to the suitability of the DCFHDA probe for measuring intracellular ROS. Some researchers, including Forman et al., 2015, have highlighted that DCFHDA is not reliable for measuring intracellular H_2_O_2_ [55]. Though this is acknowledged, Tarpey and Fridovich, 2001, have suggested that instead of employing the DCFHDA probe as an indicator for H_2_O_2_, the probe may be employed as qualitative marker for oxidative stress in cells [56]. Thus, in this study, the DCFHDA probe was used generally, to investigate the oxidative stress ameliorating properties of AE by its ROS scavenging activity in *C. elegans*.

Secondly, Gruber et al., 2009 raised interesting but valid concerns on performing survival assays in *C. elegans* studies in a blindfolded manner [57]. In this study, we depended on assay replications and the exploration of a variety of assays to confirm stated biological activities of AE. This study also utilized one worm per repeat (making three worms in total per group) to determine how AE affects the reproduction capacity of a single worm.

Additionally, O’Rourke et al., 2009 have raised the concern that the commonly used Nile red staining is not suitable for determining total lipid content in *C. elegans* [58]. Recently, Escorcia et al. 2018 highlighted that, though more reliable methods like high performance liquid chromatography-mass spectrometry (HPLC-MS), gas chromatography-mass spectrometry (GC-MS), and coherent anti-stokes Raman scattering (CARS) microscopy exist, the use of Nile red and Oil red O assays are effective in determining the general lipid content of worms [59]. In this study, Nile red assay was employed to give an idea of the general lipid reducing ability of AE in *C. elegans*. Extensive research has already been conducted by Sogo et al., 2021 (from our laboratory) on the lipolytic effect of AE on 3T3-L1 adipocytes where Oil Red O (recommended by O’Rourke et al., 2009) was employed [12].

Finally, Saul et al., 2021 also raised an interesting concern about the use of younger worms in ageing studies. They suggested that older nematodes should be used when health span/ageing is a focus of the study [60]. Though this is logical, we have observed that handling aged *C. elegans* is challenging; plates are prone to contamination and preparing aged worms for assay can also stress the worms. Because these worms are fragile, they are much more prone to quick death by handling, transfer, or mechanical assays. These were validated by Herndon et al., 2017. According to Herndon et al., 2017, ‘Generalized physical deterioration may disrupt bodily functions to a lethal extent’ [61]. Nonetheless, several studies that used young nematodes still obtained great results that confirmed their hypothesis [62].

As research advances, so do research methods. Most research methods have pros and cons. Thus, we recommend that hypotheses are proven with experiment repetitions and a combination of several assays.

## 4. Materials and Methods

### 4.1. AE Preparation and Other Reagents

Amber was crushed, powdered, and extracted twice with 50% ethanol at a temperature of 40 °C for 1 h. The filtrate was then freeze dried to obtain a powder. Amber extract was then dissolved in Dimethyl sulfoxide (DMSO; Kanto Chemical Co., Tokyo, Japan) at a stock concentration of 100 mg/mL, stored at −80 °C and supplied by Kohaku Biotechnology Co., Ltd., Tsukuba, Japan.

For treatment, AE was mixed with *E. coli* OP50 to obtain respective final concentrations of 5, 25 and 50 µg/mL at 5% DMSO and spread on Nematode Growth Medium (NGM) plates. For a control, 5% DMSO was employed.

5′-Fluorodeoxyuridine (FUDR), Ampicillin Sodium salt, Dichlorofluorescin diacetate (DCFHDA), Epigallocatechin gallate (EGCG) and Nile red were obtained from FUJIFILM Wako Pure Chemical Industries, Ltd., Osaka, Japan. Cortisone and Hydrogen peroxide were purchased from Sigma-Aldrich, St. Louis, MO, USA. Additionally, 2,2-Diphenyl-1-picrylhydrazyl (DPPH) was also purchased from Cayman Chemical Company, Ann Arbor, MI, USA.

### 4.2. C. elegans Strains, Culture and Maintenance

*C. elegans* strains; N2 Bristol (wild type), TJ356 (zIs356 (daf-16p::daf-16a/b::GFP + rol-6 (su1006))), TJ375 (gpIs1 (hsp- 16.2p::GFP)), CF1553 (muIs84 ((pAD76) sod-3p::GFP + rol-6(su1006))), CL2166 (dvIs19 ((pAF15) gst-4p::GFP::NLS) III), CB1370 (*daf-2* (*e1370*)), GR1307 (*daf-16* (*mgDf50*) I) and *hsf-1* (*sy441*) were obtained from the Caenorhabditis Genetic Centre (CGC), University of Minnesota, Minneapolis, MN, USA. *skn-1* (*tm4241*) was obtained from the National BioResource Project (NBRP), Tokyo Women’s Medical University, Japan.

Worms were cultured on Nematode Growth Medium (NGM: 1.7% agar, 25 mM potassium phosphate, pH 6.0, 50 mM NaCl, 2.5 µg/mL peptone, 5 µg/mL cholesterol, 1 mM MgSO4, 1 mM CaCl2) and seeded with *Escherichia coli* (*E. coli*) OP50 at 20 °C. Worms were age-synchronized to obtain offspring of the same growth stage by bleaching with alkaline hypochlorite (NaClO: Haiter, KAO, Tokyo, Japan).

### 4.3. Longevity Assay

A total of 30 age-synchronized L1 worms were cultured on NGM-OP50 plates, containing 100 mg/mL Ampicillin [63] at 20 °C for 96 h (day 4). FUDR was added to all plates on days 2, 4 and 6 after L1 worms were obtained, to seize worm reproduction and egg-hatching. Ampicillin was used to prevent plate contamination by foreign bacteria. The worms were transferred with a platinum wire (picker) onto OP50-only (negative control), 5% DMSO (control) and AE (5, 25 and 50 µg/mL) plates on the first day of adulthood, herein marked as day 0. Survival rate was determined every 2–3 days by scoring worms dead or alive and transferring alive worms to new plates. The experiment was conducted in triplicate.

### 4.4. Health Span Assays

#### 4.4.1. Body Length Determination

Age-synchronized N2 L1 worms were cultured on NGM-OP50 plates with DMSO and AE (5, 25 and 50 µg/mL) at 20 °C for 96 h. Worms were washed off the plates and fixed with 10% ethanol (Kanto Chemical Co., Inc, Tokyo, Japan). Images of 30 worms were captured with a BZ-8000 fluorescence microscope (Keyence, Osaka, Japan) at a magnification of 10× and their body lengths determined with ImageJ software (National Institutes of Health, Bethesda, MD, USA).

#### 4.4.2. Reproduction Capacity Assay

Age-synchronized N2 worms were cultured on NGM-OP50 plates for 72 h. One L4 stage worm each was transferred to DMSO and AE (5, 25 and 50 µg/mL) plates and cultured for 5 days. The number of eggs laid and offspring hatched were counted and scored daily before parent worms were transferred onto new treated and non-treated plates. The reproduction capacity of each worm in each group was determined and graphed.

#### 4.4.3. Motility Assay

Age-synchronized N2 L1 worms were cultured on NGM-OP50 plates with DMSO and AE (5, 25 and 50 µg/mL at 20 °C for 96 h). The speed of movement of each worm was counted in 15 s by a hand counter on day 0 (first day of adulthood), 3, 6 and 9. The movement of the head from right to left in one round was considered 1 movement. 10 worms from each group were counted on each day and the percentage motility determined. Experiments were conducted in triplicate.

### 4.5. Stress Survival Assays

#### 4.5.1. Heat Stress Survival

A total of 30 age-synchronized N2 L1 worms were cultured at 20 °C on NGM-OP50 plates treated with or without AE (50 µg/mL). After 72 h, worms were then exposed to heat shock at 35 °C for 7 h. A recovery time of 1 hr was allowed and the worms were incubated again on treated and non-treated plates at 20 °C. Their survival rate was monitored every 2 days until no worm remained alive. Percentage survival rate was then determined. This assay was conducted according to that by Kim et al., 2019 with slight modifications [37].

#### 4.5.2. Oxidative Stress Survival

Age-synchronized N2 wild-type worms, *daf-2* (*e1370*), *daf-16* (*mgDf50*), *skn-1* (*tm4241*) and *hsf-1* (*sy441*) mutants were cultured at 20 °C on NGM-OP50 plates treated with or without AE (50 µg/mL) for 96 h. On the first day of adulthood, worms were singly transferred into a 48-well plate containing 0.1% hydrogen peroxide (H₂O₂). Twelve worms/group were employed in the study and the survival rates of the worms were determined every hour until none remained alive. The percentage survival rates were then calculated and graphed. Experiments were conducted at least three times.

### 4.6. Biochemical Analysis

#### 4.6.1. Cortisol Determination

The effect of AE on the regulation of cortisol in N2 wild type worms was studied using Enzyme Linked Immunosorbent Assay (ELISA). Approximately 3000 age-synchronized L1 worms were cultured on AE-treated (50 µg/mL) and non-treated plates at 20 °C for 96 h. For cortisol levels determination, the study was first designed in our laboratory following the 11 beta hydroxysteroid dehydrogenase type 1 (11β-HSD 1) pathway where cortisone is converted to cortisol. Here, in addition to worm food, worms were cultured with (Exo, DMSO and AE (50 µg/mL) groups) and without (Endo group) 17.5 mM cortisone as a substrate. After incubation, worms were homogenized, and the lysate collected for further analysis. Protein analysis was conducted on lysate using a Cortisol ELISA kit (Cayman Chemical Company, Ann Arbor, MI, USA) according to manufacturer’s instructions. Endo represents endogenous and Exo represents exogenous cortisol production.

#### 4.6.2. Oxytocin and Dopamine Determination

Approximately 3000 age-synchronized L1 worms were cultured on AE-treated and non-treated plates at 20 °C for 96 h. Worms were then washed off the plates, homogenized and protein quantification was conducted using an Oxytocin ELISA kit (Cayman Chemical Company, Ann Arbor, MI, USA) and Dopamine ELISA kit (ImmuSmol, Bordeaux, France) according to manufacturer’s instructions. Experiments were conducted in four replicates and at three independent times.

### 4.7. Fat Accumulation Determination

A Nile red assay was employed to study the effects of AE on fat accumulation in N2 wild type worms. Age-synchronized N2 L1 worms were cultured on NGM-OP50 plates with DMSO and AE (50 µg/mL) at 20 °C for 96 h. Worms were washed off, paralyzed with 4% Paraformaldehyde (FUJIFILM Wako Pure Chemical Industries, Ltd., Osaka, Japan) in a cold room for 2 h and stained with 1 µg/mL Nile red solution for 10 min. Worms were mounted on a BZ-8000 fluorescence microscope at a magnification of 10× and the images of 30 worms were captured for body fat analysis with ImageJ software. The experiment was conducted at least three times [64].

### 4.8. Intracellular Reactive Oxygen Species (ROS) Determination

Age-synchronized L1 worms were cultured on DMSO and AE (50 µg/mL) plates at 20 °C for 96 h. Twenty worms from each AE-treated and non-treated group were transferred into a 96-well plates with 100 µL S-basal (0.01 mM cholesterol, 100 mM NaCl, and 50 mM potassium phosphate, pH 6.0) solution. 100 µL of 50 µM DCFHDA solution was added to obtain a final DCFHDA concentration of 25 µM. The plate was then incubated for 1 h in the dark. Using a fluorescence microplate reader (Biotek, Synergy H1, Tokyo, Japan), the fluorescence intensity was recorded at excitation and emission wavelengths of 485 nm and 535 nm, respectively. The experiment was conducted three different times.

### 4.9. DPPH Assay

To determine the in vitro antioxidant ability of AE, a 2,2-Diphenyl–1–picrylhydrazyl (DPPH) assay was conducted. Epigallocatechin gallate (EGCG) was used as a reference sample. An amount of 100 µL of 0.2 mM DPPH in DMSO was prepared and added to 100 µL each of 50 µg/mL AE and EGCG (5 µg/mL) in a 96-well plate. The solutions were incubated in the dark for 30 min and the absorbance was determined at 517 nm. The percentage scavenging activity was calculated according to Wang et al., 2020 as: % Scavenging effect = (1 − A1/A0) × 100) [35], where A1 is the absorbance of samples and A0 is the absorbance of blank (DPPH and DMSO). The experiment was conducted in triplicate.

### 4.10. Gene Expression

#### 4.10.1. Green Fluorescence Protein (GFP)

Synchronized L1 worms were cultured on DMSO or AE (50 µg/mL) plates at 20 °C. TJ356 worms that expressed DAF-16::GFP subcellular localization were cultured for 96 h. Worms were washed off the plates with S-basal and paralyzed with 10% ethanol. Images of 30 worms were then captured with a BZ-8000 fluorescence microscope at a magnification of 10× and the distribution of DAF-16 transcription factor was categorized into cytosolic, intermediate and nucleus. The distribution was counted and graphed in percentage. TJ375 worms expressing HSP-16.2::GFP were cultured for 72 h and then exposed to heat shock at 35 °C for 2 h. A recovery time of approximately 18 h was allowed. Worms were then washed, paralyzed, and imaged as above. The fluorescence intensity was determined for 30 worms after analysis with ImageJ software. This experiment was conducted according to that by Duangjan et al., 2019 with slight modifications [29]. CF1553 and CL2166 worms expressing SOD-3::GFP and GST-4::GFP, respectively, were cultured for 72 h and 48 h, respectively. Worms were washed, paralyzed and imaged as above. Fluorescence intensity of 30 worms from each strain was determined after analysis with ImageJ software. All experiments were conducted independently at least three times.

#### 4.10.2. Real Time Quantitative PCR

Approximately 500 age-synchronized N2 L1 worms were cultured on NGM-OP50 plates containing DMSO or AE (50 µg/mL) at 20 °C for 96 h. Total RNA was extracted from homogenized worms using Trizol reagent. Gel electrophoresis was conducted to ascertain the quality of the extracted RNA [22]. Subsequently, complementary DNA (cDNA) was synthesized using a PrimeScript Reagent Kit (Takara Biotechnology Inc., Shiga, Japan) and a real time quantitative PCR was conducted using Thunderbird SYBR Green Mix (Toyobo, Co., Ltd., Osaka, Japan) together with gene primer sequences listed in Appendix A. PCR products were checked via melting curve analysis and the efficiency of primers was assessed by evaluating melting/dissociation curves of primers from the experiment conducted. Relative mRNA expression levels were calculated using the 2^–∆∆Ct^ method. Y45F10D.4 was used as the housekeeping gene [65].

### 4.11. Statistical Analysis

For statistical analysis, at least three replicates were conducted. All graphs were generated in GraphPad prism 8.0. For survival curves, log-rank (Mantel-Cox) tests were used for analysis. Data were assessed using Student’s *t*-test for two groups and Tukey’s multiple comparison test, one-way/two-way ANOVA for three or more groups where necessary. *p* values less than 0.05 were considered statistically significant. Data were expressed as the mean ± SD.

## 5. Conclusions

The findings from this study provide scientific evidence to support the folkloric use of amber as a novel stress buffering and longevity-enhancing nutraceutical. Collectively, these findings suggest that AE exhibited stress-relieving properties which led to stress resistance and longevity. AE did not cause developmental or physical damage to *C. elegans*. Rather, muscle activity and stress resistance improved, in addition to the balance of specific stress hormones. It can be inferred that DAF-16, SKN-1 and HSF-1, through the IIS pathway and stress hormone regulation, may be responsible for the AE-mediated stress buffering and longevity effect. These novel discoveries have shown that amber is a potent nutraceutical for reducing stress and its damaging effects, which is essential for maintaining quality-of-life and a long life.

## Figures and Tables

**Figure 1 molecules-27-03858-f001:**
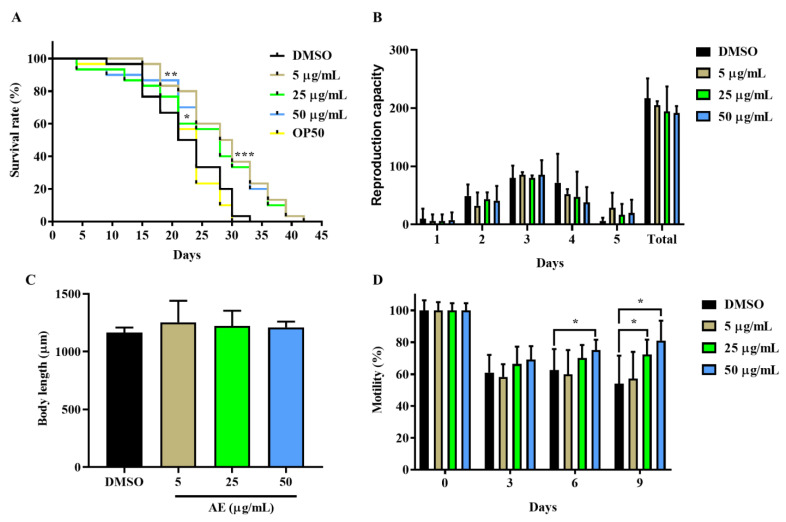
Effect of AE on lifespan and health span of wild type *C. elegans*. Worms were cultured at 20 °C on NGM-OP50 plates containing DMSO or AE (5, 25 and 50 µg/mL). (**A**) Lifespan of worms treated with or without AE (*n* = 30 worms/group). Statistical differences compared to control (DMSO) were considered significant at * *p* < 0.05, ** *p* < 0.01, *** *p* < 0.005 by log-rank test. (**B**) Reproduction ability of AE-treated and non-treated worms. The number of worms per group was one and the data of three independent experiments were represented. *n* = 3. (**C**) Body length determination of worms fed with or without AE for 96 h (*n* = 20 worms/group). (**D**) Motility of AE-treated and non-treated worms (*n* = 10 worms/group). Statistical differences compared to control (DMSO) were considered significant at * *p* < 0.05 by two-way ANOVA. Data were represented by mean ± SD. Experiments were performed in triplicate determinations.

**Figure 2 molecules-27-03858-f002:**
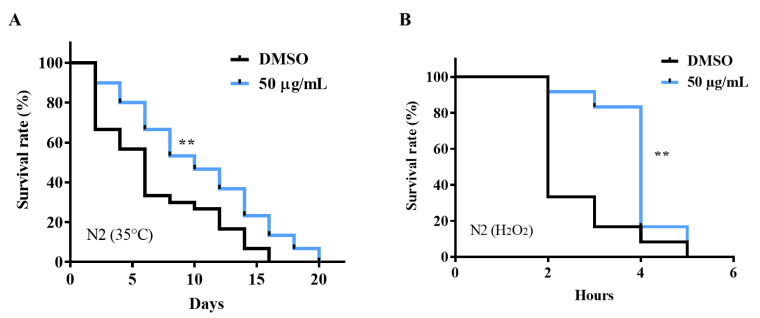
Effect of AE treatment on stress tolerance of wild type *C. elegans*. Worms were cultured on NGM-OP50 plates containing DMSO or AE (50 µg/mL) at 20 °C for 96 h. (**A**) Heat stress survival of worms exposed to heat at 35 °C for 7 h (*n* = 30 worms/group) (**B**) Oxidative stress survival of worms exposed to 0.1% hydrogen peroxide (*n* = 12 worms/group). Statistical differences compared to control (DMSO) were considered significant at ** *p* < 0.01 by log-rank test. Data shown are representative of triplicate determinations.

**Figure 3 molecules-27-03858-f003:**
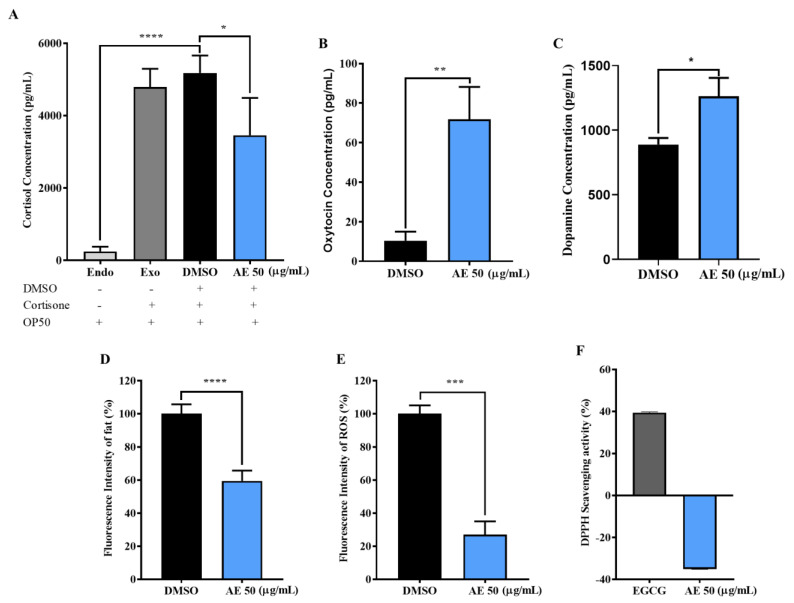
Effect of AE treatment on stress biomarkers in wild type *C. elegans*. Worms were cultured at 20 °C on NGM-OP50 plates, treated with or without AE (50 µg/mL) for 96 h. (**A**) Cortisol production was assessed through the 11 beta hydroxysteroid dehydrogenase pathway and its quantity in worm homogenate determined by ELISA (*n* ≈ 3000 worms). Statistical differences compared to control (DMSO) were considered significant at * *p* < 0.05, **** *p* < 0.0001 by one-way ANOVA. Endo represents endogenous cortisol production. Exo represents exogenous cortisol production (cortisone addition). (**B**) Oxytocin levels in worm homogenate were quantified by ELISA (*n* ≈ 3000 worms), (**C**) Dopamine levels in worm homogenate were quantified by ELISA (*n* ≈ 3000 worms), (**D**) Fat accumulation was determined by Nile red staining (*n* = 10 worms/group), (**E**) Intracellular ROS accumulation was determined by DCFHDA assay after 1 hr incubation time (*n* = 20 worms/group) and (**F**) In vitro ROS determination by DPPH assay after 30 min incubation time. Statistical differences compared to control (DMSO) were considered significant at * *p* < 0.05, ** *p* < 0.01, *** *p* < 0.005 and **** *p* < 0.0001 by Student’s *t*-test. Data were represented as mean ± SD. Experiments were performed in triplicate determinations.

**Figure 4 molecules-27-03858-f004:**
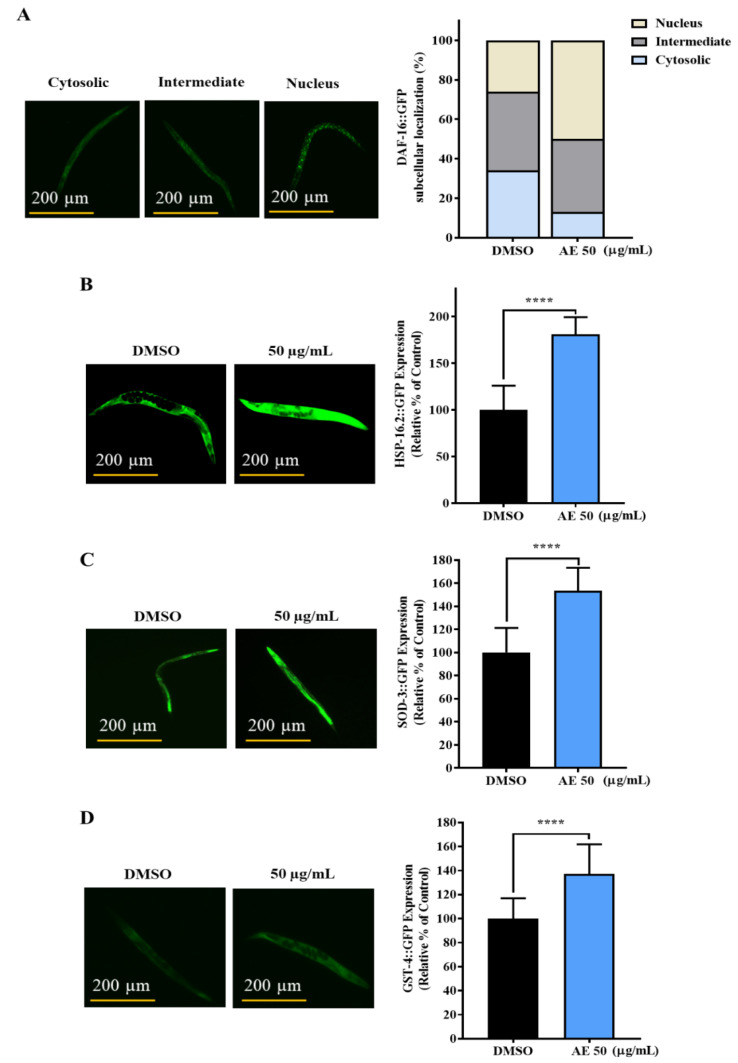
Effects of AE on DAF-16 cellular translocation and the expression of HSP-16.2, SOD-3 and GST-4 in GFP-tagged worms. (**A**) Subcellular localization of DAF-16::GFP worms treated with or without AE (50 µg/mL) for 96 h (*n* = 30 worms/group). Subcellular localization was categorized into Cytosolic, Nucleus and Intermediate (between cytosolic and nucleus). (**B**) HSP-16.2::GFP worms treated or not treated with AE were cultured for 72 h and exposed to heat shock at 35 °C for 2 h (*n* = 30 worms/group). (**C**) SOD-3::GFP and (**D**) GST-4::GFP AE-treated and non-treated worms were cultured for 72 h (*n* = 30 worms/group). Statistical differences compared to control (DMSO) were considered significant at **** *p* < 0.001 by Student’s *t*-test. Data were represented by mean ± SD. Experiments were performed in triplicate determinations and representatives shown. Scale bar for GFP analysis = 200 µm.

**Figure 5 molecules-27-03858-f005:**
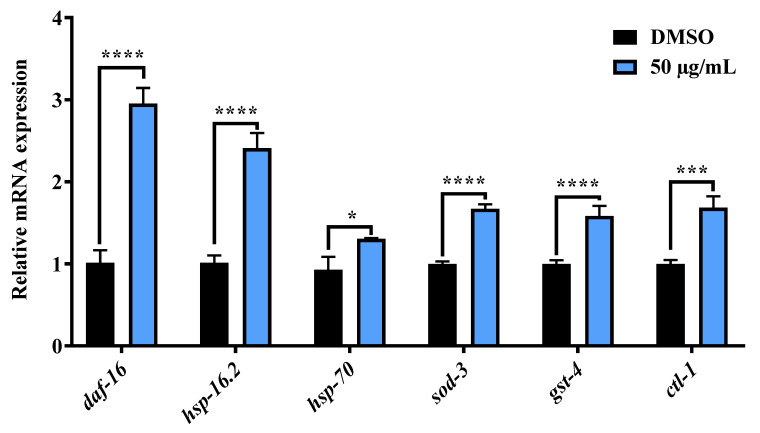
Effects of AE on the mRNA expression *daf-16* and its target genes in wild type *C. elegans*. Worms were cultured at 20 °C on NGM-OP50 plates, treated with or without AE (50 µg/mL) for 96 h (*n* ≈ 500 worms/group). Statistical differences compared to control (DMSO) were considered significant at * *p* < 0.05, *** *p* < 0.005 and **** *p* < 0.0001 by multiple t tests. Data were represented by mean ± SD. Experiments were performed in triplicate determinations.

**Figure 6 molecules-27-03858-f006:**
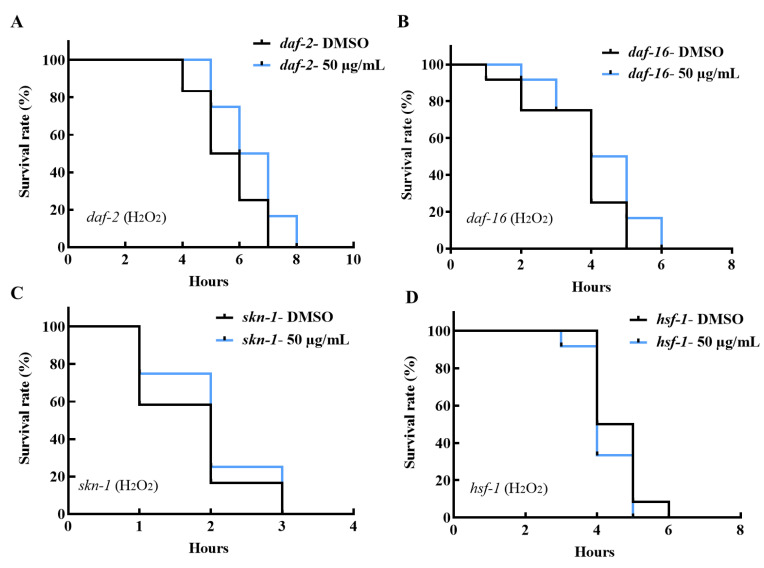
Effect of AE on the oxidative stress survival of *C. elegans* mutants. Worms were cultured at 20 °C on NGM-OP50 plates, treated with or without AE (50 µg/mL) for 96 h. (**A**) *daf-2* (*e1370*) mutants, (**B**) *daf-16* (*mgDf50*) mutants, (**C**) *skn-1* (*tm4241*) mutants and (**D**) *hsf-1* (*sy441*) mutants oxidative stress survival (*n* = 12 worms/group). Statistical significances of survival curves were assessed by log-rank test. Experiments were performed in triplicate determinations and representative curves shown.

## Data Availability

Data is contained within the article or Appendix A.

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
