# Peer review of "Stress Buffering and Longevity Effects of Amber Extract on Caenorhabditis elegans (C. elegans)"

_molecules, 2022, doi:10.3390/molecules27123858_

Round 1
Reviewer 1 Report
The article was improved according to my suggestions. Thus, I would suggest to accept it in its present form.
Author Response
Authors` Response:
We are grateful to the reviewer for the immense contributions towards the improvement of our paper.

Reviewer 2 Report
Even though the manuscript was improved by adding some bibliographic information on the content of phytochemicals that has been reported in amber, I insist that they need to make some analytical determinations on the phytocompounds in the extracts that they used. This will help to correlate the effects observed in their trials with the compounds containing the amber extracts used and to determine similarities/differences with the phytocompounds reported in the literature.
Author Response
Authors` Response:
We are thankful to the reviewer for the valuable suggestions given. Indeed, we agree that identifying the phytochemicals present in our Amber will help correlate the effect observed in the trials with our compounds. Though this phase of our research work is still ongoing, we wish to protect our current data from theft or plagiarism by publishing it.
The suggestions given would be helpful in our next project where we focus on the individual activities of Amber extract in relation to the phytochemicals identified from our research and in comparison to that in literature.

Round 2
Reviewer 2 Report
It is a good job, I hope they complete it in the future, as they say, with the analysis of the phytochemicals present in the amber extracts used.
This manuscript is a resubmission of an earlier submission. The following is a list of the peer review reports and author responses from that submission.
Round 1
Reviewer 1 Report
The paper entitled Stress Buffering and Longevity Effects of Amber extract on Caenorhabditis elegans (C. elegans) with the aim of investigating the effect of an amber extract (EA) on the survival of C. elegans under normal conditions and when exposed to stress . The authors also studied the effect of EA on the regulation of biomarkers of stress; levels of hormones/neurotransmitters, ROS and fat accumulation. Expression of longevity and stress regulatory genes was also monitored at GFP-fused reporters and at mRNA levels. Molecular pathways that may be involved in AE-mediated activity were determined using loss-of-function strains (mutants) of C. elegans.
The paper is interesting, it is well written, it is clear, however, they used a crude extract in which the main phytochemicals present in that extract (phenolic acids, flavonones, terpenoids, etc.) were not identified or quantified, which I consider essential to help understand and /or correlate the observed effects with the compounds in the extract.
I suggest carrying out the missing tests in the extract on the composition of the main phytochemicals to include it in the paper. As it currently stands, I do not consider it appropriate for publication in the journal Molecules.
Author Response
We are thankful to the reviewer for the suggestions and considering our manuscript interesting, clear and well written.
We apologize for not extensively talking about Amber and its components in the introduction. Infact, several components and active ingredients of different types of Amber have been elucidated by Maruyama M. et al. (Maruyama et al., Fototerapia, 2018; Anti-allergy activities of Kuji amber extract and kujigamberol). Specifically, Baltic Amber has been reported to contain agathic acid, primaric acid and abietic acid. (Maruyama et al., Fototerapia, 2018; Anti-allergy activities of Kuji amber extract and kujigamberol, Antibacterial activity and GC-MS analysis of baltic amber against pathogenic
Bacteria; https://faculty.uobasrah.edu.iq/uploads/publications/1599035651.pdf)
Currently, we are working on identifying more of the unknown components of Amber extract and only wanted to publish the current data available on Amber extract.
We would therefore like to effect changes in the manuscript by explaining further the components of Amber extract.
Please find below, the changes in the manuscript.
`The diterpenoids agathic acid, primaric acid and abietic acid have been discovered as the major components of Baltic Amber [15,16]`.
Reviewer 2 Report
I think that your work should not focuse only on only one product and show the influence of several anti-oxidants without to be so deep on the biological analyse. Amber is perhaps not the best of the product and more available products can perhaps more active.
Author Response
We thank the reviewer for this comment. Infact, Amber is a promising phytochemical with several health benefits. We believe that it is necessary to conduct in-dept research on it including its biological activities to provide evidence for its therapeutic effect. At present, this research is a part of a bigger project comprising of a team of researchers focusing on the elucidation of the bioactive components and activities of Amber extract. In doing so, we contribute to saving lives and providing quality and healthy living to humanity.
Reviewer 3 Report
The article “Stress Buffering and Longevity Effects of Amber extract on Caenorhabditis elegans (C. elegans)” submitted to “Molecules” provides interesting insights into the health- and lifespan promoting action of Amber extract in the model organism C. elegans and its molecular and metabolic background mechanisms. The main focus of this study was placed on stress resistance, whereby the introduction gives a good summary of the importance of stress for healthspan. The uncovering and understanding of natural compounds with anti-ageing and pro-longevity capacities is of general interest and well-selected methods were used to determine the effects on lifespan and healthspan in this study. Furthermore, several interesting genes were targeted via mutant strain analyses as well as in a qPCR assay. However, several issues should be addressed prior publication:
- Please include one table (or several separate tables) in the attachment for all survival and lifespan experiments indicating at least the logrank p-values, the mean lifespan/mean survival, as well as the number of observed worms per group after censoring.
- Endogenous oxidative stress was determined by the H2DCFDA assay. Despite its frequent use, the H2DCFDA assay was found to be unsuitable to measure intracellular ROS in elegans, which was affirmed by the editorial board of the journal “Free Radical Biology and Medicine” and by Labuschagne and Brenkman (2013) (Labuschagne CF, Brenkman AB (2013) Current methods in quantifying ROS and oxidative damage in Caenorhabditis elegans and other model organism of aging. Ageing Res Rev 12:918–930.) as well as Dikalov and Harrison (2014) (Dikalov SI, Harrison DG (2014) Methods for detection of mitochondrial and cellular reactive oxygen species. Antioxid Redox Signal 20:372–382.). Thus, the interpretation of the results should be done very carefully. Furthermore, the problems by using this assay should be discussed.
- Plant polyphenols and extracts were often shown to have diverse antibacterial effects. By adding these compounds to the alive feeding bacteria (OP50), a compound-bacteria interaction cannot be excluded (discussed for instance in Saul et al., 2021, Health and longevity studies in C. elegans: the “healthy worm database” reveals strengths, weaknesses and gaps of test compound-based studies. Biogerontology 22, 215–236.). The antimicrobial activity could inhibit bacterial proliferation, which in turn could, for instance, reduce the harmful intestinal accumulation of bacteria and produce life- and healthspan benefits. Therefore, to exclude such a mechanism, you could test whether AE shows any antibacterial impact against E. coli OP50 (the bacterial growth could be determined by frequently measuring the optical density during exposure to the extract in the used concentrations). Alternatively, a suitable paper with a determination of Amber-E. coli interactions could be cited.
- Results, Figure 1B: You indicated that you counted the brood size daily, but Fig. 1B only shows the total number of offspring. Please show, in addition, the daily number of counted offspring and mention, whether the timing (progeny per day) of reproduction changed or not. For instance, in Sayed et al., (2021, Enhanced healthspan in Caenorhabditis elegans treated with extracts from the Traditional Chinese Medicine plants Cuscuta chinensis Lam. and Eucommia ulmoides Oliv. Frontiers in Pharmacology 12, 18.) it was shown, that the extract didn’t change the total offspring, but the timing of reproduction changed when comparing the daily offspring numbers. This could be an interesting additional information.
- Results, Figure 1C: Please indicate the real lengths (in µm) of worms instead of percentage values.
- Results, line 107: Why is it surprising that the motility was enhanced by Amber? You expected that AE is improving the healthspan, and motility is part of healthspan.
- Results, line 115: You only used 30 worms per group for the lifespan assay (and also for the stress survival assays). This seems to be pretty little. Furthermore, you did not perform the experiments in a blinded manner. In manually performed C. elegans analyses, especially lifespan or stress resistance, researchers need to decide whether a worm is dead or alive by visual judgement. A certain expectation, such as an expected life prolonging ability of a substance, can have a substantial influence on this decision. Gruber et al. (2009) (Deceptively simple but simply deceptive–Caenorhabditis elegans lifespan studies: considerations for aging and antioxidant effects. FEBS Lett 583:3377–3387.) explained the operator bias in detail and suggest blinding and randomization especially for all survival studies in elegans. Furthermore, they also discuss the problem of studies with low numbers of worms. Both limits the objectivity of the presented results and this problem should be discussed in the article. In addition, please indicate in the method section whether worms that crawl off plates, die of injury or suffer internal hatching of larvae were censored.
- Results, line 118: Also for the reproduction assay, the number of monitored worms is extremely low. Please discuss that limitation in the article.
- Results, Figure 3D: In 2009, O'Rourke et al. ( C. elegans major fats are stored in vesicles distinct from lysosome-related organelles. Cell Metab 10:430–435) could show that the commonly used Nile red staining is not suitable to determine the total lipid content in C. elegans. This needs to be discussed in your paper.
- The phenotypic measurements and also the gene expression study were performed in quite young nematodes. As discussed in Saul et al., 2021 (Health and longevity studies in C. elegans: the “healthy worm database” reveals strengths, weaknesses and gaps of test compound-based studies. Biogerontology 22, 215–236.) measurements should be rather performed in older nematodes, when healthspan/ageing is a focus of the study. Results from studies in young and aged nematodes can differ to a great extent and this should be at least discussed.
- Methods: Please clearly indicate when the AE-treatments started (at the egg-stage, L1, L4 or the first day of adulthood?)
- Methods: Please indicate the used microscope magnifications in the chapters 4.4.1, 4.7, and 4.10.1.
- Methods, line 459: Why do you talk about the worms used for body length assay here? Please rewrite this sentence more clear
- Methods, line 499 ff: Please indicate whether the PCR products were checked via gel electrophoresis and melting curve analysis. Furthermore, also specify the used annealing temperatures (together with the primer sequences), product sizes and the determined primer pair efficiencies in a separated table (as attachment). If you didn’t determine primer efficiencies, this limitation needs to be mentioned/discussed. Furthermore, only one reference gene was used in this study which limits the interpretation of the results. This should be also at least discussed or re-analysed using a second reference gene.
- Methods, line 385: What do you mean with day (-2, 0, 2)? Please write that clearer.
- Methods, line 409: what do you mean with “Movement of worms were counted in 15 seconds on day …”. Did you count how many worms moved on the plate during 15 seconds or did you determine the speed of movement (if so, how?) of each worm for 15 seconds?
Author Response
We thank the reviewer for his comments, suggestions and guidance. Below is our attempt to address and implement all concerns and recommendations.
- Reviewer`s Comment:
Please include one table (or several separate tables) in the attachment for all survival and lifespan experiments indicating at least the logrank p-values, the mean lifespan/mean survival, as well as the number of observed worms per group after censoring.
Authors` Response:
In this study because a fixed number of worms (30) was used to avoid censoring (common in many C. elegans lifespan assays), attached are tables showing survival assays with number of worms per group, median lifespans and logrank p-values. We would like to state that all survival experiments were conducted in triplicate and the best dataset was presented.
- Reviewer`s Comment:
Endogenous oxidative stress was determined by the H2DCFDA assay. Despite its frequent use, the H2DCFDA assay was found to be unsuitable to measure intracellular ROS in elegans, which was affirmed by the editorial board of the journal “Free Radical Biology and Medicine” and by Labuschagne and Brenkman (2013) (Labuschagne CF, Brenkman AB (2013) Current methods in quantifying ROS and oxidative damage in Caenorhabditis elegans and other model organism of aging. Ageing Res Rev 12:918–930.) as well as Dikalov and Harrison (2014) (Dikalov SI, Harrison DG (2014) Methods for detection of mitochondrial and cellular reactive oxygen species. Antioxid Redox Signal 20:372–382.). Thus, the interpretation of the results should be done very carefully. Furthermore, the problems by using this assay should be discussed.
Authors` Response:
Employing H2DCFDA assay was useful for us to fairly have an ideal of the antioxidant ability of Amber extract. The quest to determine ROS using molecular probe has been very challenging. As stated by Labuschagne and Brenkman (2013) in Labuschagne CF, Brenkman AB (2013); Current methods in quantifying ROS and oxidative damage in Caenorhabditis elegans and other model organism of aging. Ageing Res Rev 12:918–930, `Presently, not a single method exists to get full insight into the cellular redox state at any given moment in time, which compromises the full understanding of the role of ROS and its damage to aging`. We believe that the H2DCFDA assay is commonly used because despite the challenges faced in determining ROS, this method is much more reliable than many other probes. The ideal approach is to validate results by using a combination of several assays for ROS detection and quantification as recommended by Labuschagne and Brenkman (2013) in Labuschagne CF, Brenkman AB (2013); Current methods in quantifying ROS and oxidative damage in Caenorhabditis elegans and other model organism of aging. Ageing Res Rev 12:918–930 recommends. We are presently determining, in another research, the antioxidant activities of Amber extract by employing multiple approaches as a validation. Thus, this recommendation has been included in our current/future works.
- Reviewer`s Comment:
Plant polyphenols and extracts were often shown to have diverse antibacterial effects. By adding these compounds to the alive feeding bacteria (OP50), a compound-bacteria interaction cannot be excluded (discussed for instance in Saul et al., 2021, Health and longevity studies in C. elegans: the “healthy worm database” reveals strengths, weaknesses and gaps of test compound-based studies. Biogerontology 22, 215–236.). The antimicrobial activity could inhibit bacterial proliferation, which in turn could, for instance, reduce the harmful intestinal accumulation of bacteria and produce life- and healthspan benefits. Therefore, to exclude such a mechanism, you could test whether AE shows any antibacterial impact against E. coli OP50 (the bacterial growth could be determined by frequently measuring the optical density during exposure to the extract in the used concentrations). Alternatively, a suitable paper with a determination of Amber-E. coli interactions could be cited.
Authors` Response:
Baltic Amber containing agathic acid, primaric acid and abietic acid, have been shown by Al-Tamimi, 2020 to exhibit anti-bacterial activity against E. Coli and other bacteria (Al-Tamimi et al., 2020; https://faculty.uobasrah.edu.iq/uploads/publications/1599035651.pdf). Additionally, the oleoresin of Copaiba containing agathic acid has been reported to show antibacterial activity against E. coli and many other bacteria (Leandro et al., 2012; Chemistry and Biological Activities of Terpenoids from Copaiba (Copaifera spp.) Oleoresins. Molecules 2012, 17). Xin, 2016, also highlighted that agathic acid has antibacterial effect (Xin et al, 2016; Synthesis of (−)-agathic acid and (−)-copalic acid from andrographolide via a regioselective Barton-McCombie reaction, Tetrahedron, Volume 72). This implies that the Amber extract has antibacterial effect which may have contributed to prolonging the lifespan of worms.
Kindly find the changes to the manuscript below.
`Amber, containing agathic acid, primaric acid and abietic acid has been reported to possess antibacterial activity against E. coli and other bacteria [16]. Perhaps this anti-bacterial activity of Amber contributed to lifespan extension of worms by inhibiting the proliferation and accumulation of live E. coli in worms` gut that could negatively affect health span and lifespan`.
- Reviewer`s Comment:
Results, Figure 1B: You indicated that you counted the brood size daily, but Fig. 1B only shows the total number of offspring. Please show, in addition, the daily number of counted offspring and mention, whether the timing (progeny per day) of reproduction changed or not. For instance, in Sayed et al., (2021, Enhanced healthspan in Caenorhabditis elegans treated with extracts from the Traditional Chinese Medicine plants Cuscuta chinensis Lam. and Eucommia ulmoides Oliv. Frontiers in Pharmacology 12, 18.) it was shown, that the extract didn’t change the total offspring, but the timing of reproduction changed when comparing the daily offspring numbers. This could be an interesting additional information.
Authors` Response:
Unfortunately, we found no significant differences in daily or total brood size of worms with or without treatment.
Kindly find the changes made in Fig. 1B. `It was found in Fig. 1B that AE supplementation did not significantly change both daily and overall fertility of worms`.
- Reviewer`s Comment:
Results, Figure 1C: Please indicate the real lengths (in µm) of worms instead of percentage values.
Authors` Response: 
Kindly find changes in the manuscript as shown in Fig. 1C.
- Reviewer`s Comment:
Results, line 107: Why is it surprising that the motility was enhanced by Amber? You expected that AE is improving the healthspan, and motility is part of healthspan.
Authors` Response:
We apologize for the wrong use of adverb. We modified the sentence this as shown below.
`In Fig. 1D, 25 µg/mL and 50 µg/mL AE treatment enhanced muscle movement as worms aged: day 6 and day 6 and 9 respectively`.
- Reviewer`s Comment:
Results, line 115: You only used 30 worms per group for the lifespan assay (and also for the stress survival assays). This seems to be pretty little. Furthermore, you did not perform the experiments in a blinded manner. In manually performed C. elegans analyses, especially lifespan or stress resistance, researchers need to decide whether a worm is dead or alive by visual judgement. A certain expectation, such as an expected life prolonging ability of a substance, can have a substantial influence on this decision. Gruber et al. (2009) (Deceptively simple but simply deceptive–Caenorhabditis elegans lifespan studies: considerations for aging and antioxidant effects. FEBS Lett 583:3377–3387.) explained the operator bias in detail and suggest blinding and randomization especially for all survival studies in elegans. Furthermore, they also discuss the problem of studies with low numbers of worms. Both limits the objectivity of the presented results and this problem should be discussed in the article. In addition, please indicate in the method section whether worms that crawl off plates, die of injury or suffer internal hatching of larvae were censored.
Authors` Response:
Gruber et al. (2009) (Deceptively simple but simply deceptive–Caenorhabditis elegans lifespan studies: considerations for aging and antioxidant effects. FEBS Lett 583:3377–3387.) presented an interesting but valid concern that pertain to C. elegans lifespan and survival assays. With regards to the number of worms used, ideally higher number of worms is essential when worms must be censored due to worms crawling off plates or unexpected injury causing death. However, our study avoided the circumstance of having to censor worms by spreading OP50 in the center of NGM plates so that worms do not craw off. Worms that were unfortunately injured (happens less often) or contaminated plates, prompted for a stop and exclusion of that data and a repetition of the experiment. Suffering internal larvae was avoided as worms carrying eggs were not selected for the lifespan assay. In view of this, we selected 30 worms with a triplicate determination in each group (making 90 worms) and the best data set was presented.
As research advances, so does research methods. Ideally, automated system of determining lifespan should be used to help reduce manual judgement of whether a worm is alive or dead. But the cost of these systems makes it quite impossible to avoid manual use. Thus, we depended on experiment repetition and combination of several other assays like the gene expression assay and others to validate our results.
- Reviewer`s Comment:
Results, line 118: Also for the reproduction assay, the number of monitored worms is extremely low. Please discuss that limitation in the article.
Authors` Response:
This assay was conducted according to that by Wang, 2020; Wang J, Deng N, Wang H, et al. Effects of Orange Extracts on Longevity, Healthspan, and Stress Resistance in Caenorhabditis elegans. Molecules. 2020. In conducting this assay, the experiment was repeated 3 times and the results of only one worm was shown to throw more light on the reproduction capacity of an individual worm with or without Amber extract treatment.
- Reviewer`s Comment:
Results, Figure 3D: In 2009, O'Rourke et al. ( C. elegans major fats are stored in vesicles distinct from lysosome-related organelles. Cell Metab 10:430–435) could show that the commonly used Nile red staining is not suitable to determine the total lipid content in C. elegans. This needs to be discussed in your paper.
Authors` Response:
Recently in 2018, Escorcia et al. (Quantification of Lipid Abundance and Evaluation of Lipid Distribution in Caenorhabditis elegans by Nile Red and Oil Red O Staining. Journal of Visualized Experiments, (133)) highlighted that though more reliable methods like high performance liquid chromatography-mass spectrometry (HPLC-MS), gas chromatography-mass spectrometry (GC-MS), and coherent anti-stokes Raman scattering (CARS) microscopy exist, the use of Nile Red and Oil Red O assays are effective in determining the general lipid content of worms. In this study, we sort to determine the general lipid reducing ability of Amber extract using Nile red assay to support our story. We look forward to exploring further the lipid reducing abilities of Amber extract in C. elegans using Oil red O assay; a method recommended by O'Rourke et al. (C. elegans major fats are stored in vesicles distinct from lysosome-related organelles. Cell Metab 10:430–435) and several other approaches.
- Reviewer`s Comment:
The phenotypic measurements and also the gene expression study were performed in quite young nematodes. As discussed in Saul et al., 2021 (Health and longevity studies in C. elegans: the “healthy worm database” reveals strengths, weaknesses and gaps of test compound-based studies. Biogerontology 22, 215–236.) measurements should be rather performed in older nematodes, when healthspan/ageing is a focus of the study. Results from studies in young and aged nematodes can differ to a great extent and this should be at least discussed.
Authors` Response:
We observed that several studies that used young nematodes still obtained great results that confirmed their hypothesis (Gusarov et al, 2021; Dietary thiols accelerate aging of C. elegans. Nat Commun 12, 4336).
Since C. elegans must be transferred to new OP50 plates about 3-4 days as maintenance, the plates are also prone to contamination which may affect data obtained from aged groups.
In our experiments we focused on being much more consistent with the aged group to fairly compare the effects of AE.
Although Saul et el, raised interesting concerns, we have observed that handling aged C. elegans is challenging. Preparing worms for assay can render them stressed. Also, aged worms are much more prone to early death with little transfers or mechanical assays. This was validated by Herndon et al., 2017 in a book by Olsen, Anders; Gill, Matthew S. (2017)- Healthy Ageing and Longevity; Ageing: Lessons from C. elegans; Effects of Ageing on the Basic Biology and Anatomy of C. elegans.,10.1007/978-3-319-44703-2 (Chapter 2), 9–39. doi:10.1007/978-3-319-44703-2_2). Also, According to Herndon et al, `Generalized physical deterioration may disrupt bodily functions to a lethal extent`. We believe that these challenges make young nematodes preferable.
- Reviewer`s Comment:
Methods: Please clearly indicate when the AE-treatments started (at the egg-stage, L1, L4 or the first day of adulthood?)
Authors` Response:
The various changes have been made in the manuscript.
- Reviewer`s Comment:
Methods: Please indicate the used microscope magnifications in the chapters 4.4.1, 4.7, and 4.10.1.
Authors` Response:
Microscope magnifications used is 10x.
Changes have been made in the manuscript.
- Reviewer`s Comment:
Methods, line 459: Why do you talk about the worms used for body length assay here? Please rewrite this sentence more clear
Authors` Response:
We are sorry for this error. Changes have been made in the manuscript as below.
…the images of 30 worms were captured for body fat analysis with ImageJ software.
- Reviewer`s Comment:
Methods, line 499 ff: Please indicate whether the PCR products were checked via gel electrophoresis and melting curve analysis. Furthermore, also specify the used annealing temperatures (together with the primer sequences), product sizes and the determined primer pair efficiencies in a separated table (as attachment). If you didn’t determine primer efficiencies, this limitation needs to be mentioned/discussed. Furthermore, only one reference gene was used in this study which limits the interpretation of the results. This should be also at least discussed or re-analysed using a second reference gene.
Authors` Response:
We checked RNA quality by gel electrophoresis stated in line 502 that, ‘Gel electrophoresis was conducted to ascertain the quality of the extracted RNA [20]`. PCR products were checked via melting curve analysis. The efficiency of primers was assessed by evaluating melting/dissociation curves of primers from the experiment conducted.
According to Hoogewijs et al, 2008 (Selection and validation of a set of reliable reference genes for quantitative sod gene expression analysis in C. elegans. BMC Molecular Biol 9, 9 (2008)), Y45F10D.4 showed a stable expression pattern when used as a housekeeper for the expression 5 Sods. This effect was also observed by Zhang et al, 2012 in Selection of Reliable Reference Genes in Caenorhabditis elegans for Analysis of Nanotoxicity, PloS One.
Additionally, Takaya Sugawara from our lab conducted a screening of 3 reference genes; Actin, tba-1and Y45F10D.4 in his study. He found that Y45F10D.4 was more suitable as a reference gene (results shown below).
Based of these two recommendations, only Y45F10D.4 was considered as a suitable reference gene for our experiment.
Primer details and annealing temperatures have been stated in attached table (Table S1).
- Reviewer`s Comment:
Methods, line 385: What do you mean with day (-2, 0, 2)? Please write that clearer.
Authors` Response:
In this study, day 0 represents the first day of adulthood. Thus, we modified the sentence as `FUDR was added to all plates, on days 2, 4 and 6 after L1 worms were obtained, to seize worm reproduction and egg-hatching`.
- Reviewer`s Comment:
Methods, line 409: what do you mean with “Movement of worms were counted in 15 seconds on day …”. Did you count how many worms moved on the plate during 15 seconds or did you determine the speed of movement (if so, how?) of each worm for 15 seconds?
Authors` Response:
Please find changes in the manuscript as below.
The speed of movement of each worm was counted in 15 seconds by a hand counter on day 0 (first day of adulthood), 3, 6 and 9. The movement of the head from right to left in one round was considered 1 movement.
Round 2
Reviewer 1 Report
I consider it necessary to carry out the analyzes on the extracts used in order to be able to correlate them with their results and not rely only on what is reported in the bibliography. The composition of the biocomposites can vary greatly due to several factors such as the source of origin of the amber, extraction methods, solvents used, etc.
I recommend waiting for those analytical results of your amber extracts and resubmitting the paper.
Even with the little information added, in my opinion it is not ready for publication in Molecules, which is a journal with a high scientific impact.
Reviewer 2 Report
In fact the authors have done a lot of no conclusive experiment.......Do you have an action on bacteria or oxidative stress? It seems to me that the question seems not solved.
Reviewer 3 Report
The method parts in the revised version of the manuscript “Stress Buffering and Longevity Effects of Amber extract on Caenorhabditis elegans (C. elegans)” submitted to “Molecules” were improved according to my suggestions. However, the limitations of the study were only discussed in the response letter, but were not included in the manuscript. In order to follow good scientific practice, I suggest that potential concerns of a performed study should be also mentioned and discussed publicly (not only in the response letter). Therefore, the following points should be also addressed inside the manuscript (for instance, as part of an extra “limitations of the study” chapter in the discussion or as part of the respective results sections). The respective papers dealing with these issues should also be cited. Thus, the following points are still open:
- The potential problems by using the H2DCFDA assay should be discussed. Despite its frequent use, the H2DCFDA assay was found to be unsuitable to measure intracellular ROS in elegans, which was affirmed by the editorial board of the journal “Free Radical Biology and Medicine” (https://escholarship.org/content/qt62m745fk/qt62m745fk.pdf) and by Labuschagne and Brenkman (2013) (Labuschagne CF, Brenkman AB (2013) Current methods in quantifying ROS and oxidative damage in Caenorhabditis elegans and other model organism of aging. Ageing Res Rev 12:918–930.) as well as Dikalov and Harrison (2014) (Dikalov SI, Harrison DG (2014) Methods for detection of mitochondrial and cellular reactive oxygen species. Antioxid Redox Signal 20:372–382.). Thus, the interpretation of the results should be done very carefully. Furthermore, you answered that “We believe that the H2DCFDA assay is commonly used because despite the challenges faced in determining ROS, this method is much more reliable than many other probes.” Please, provide suitable citations to underline the reliability of the H2DCFDA assay.
- You did not perform the experiments in a blinded manner (which is also possible in manually performed survival experiments). Please mention and discuss the limitations of the study also according to that fact by using, for instance, Gruber et al. (2009) (Deceptively simple but simply deceptive–Caenorhabditis elegans lifespan studies: considerations for aging and antioxidant effects. FEBS Lett 583:3377–3387.).
- Please mention the limitations by using only one worm per repeat (so in total, 3 worms per group) in the reproduction assay. Smaller effects could be easily overlooked by using such a small number, especially because a trend of lower offspring-number is already visible in the treatment groups (see Fig. 1B). This should be mentioned and discussed.
- In 2009, O'Rourke et al. ( C. elegans major fats are stored in vesicles distinct from lysosome-related organelles. Cell Metab 10:430–435) could show that the commonly used Nile red staining is not suitable to determine the total lipid content in C. elegans. In your response letter, you mentioned an interesting paper (Escorcia et al. 2018), which should be also used to discuss this issue in more detail inside the manuscript.
- The phenotypic measurements and also the gene expression study were performed in quite young nematodes. As discussed in Saul et al., 2021 (Health and longevity studies in C. elegans: the “healthy worm database” reveals strengths, weaknesses and gaps of test compound-based studies. Biogerontology 22, 215–236.) measurements should be rather performed in older nematodes, when healthspan/ageing is a focus of the study. Results from studies in young and aged nematodes can differ to a great extent and this should be at least discussed. In addition to that, the citations and arguments, presented in your response letter could be integrated in the manuscript, in order to discuss this point.
- Regarding the PCR analysis: Please integrate your following responses into the manuscript:
- “PCR products were checked via melting curve analysis.” (Note: The additional checking of the PCR-products via gel electrophoresis should be done in addition in your future studies)
- “The efficiency of primers was assessed by evaluating melting/dissociation curves of primers from the experiment conducted.” (Note: please explain that in more detail. Usually, to receive the primer pair efficiencies, a dilution series will be used to create a standard curve. The resulting efficiencies are then given in percent. See for instance https://toptipbio.com/calculate-primer-efficiencies/. If you didn’t determine real primer efficiencies, this limitation needs to be mentioned.)